# PACT: Parameterized Clipping Activation for Quantized Neural Networks

## Abstract

Deep learning algorithms achieve high classification accuracy at the expense of significant computation cost. To address this cost, a number of quantization schemes have been proposed - but most of these techniques focused on quantizing weights, which are relatively smaller in size compared to activations. This paper proposes a novel quantization scheme for activations during training - that enables neural networks to work well with ultra low precision weights and activations without any significant accuracy degradation. This technique, PArameterized Clipping acTivation (PACT), uses an activation clipping parameter $\alpha$ that is optimized during training to find the right quantization scale. PACT allows quantizing activations to arbitrary bit precisions, while achieving much better accuracy relative to published state-of-the-art quantization schemes. We show, for the first time, that both weights and activations can be quantized to 4-bits of precision while still achieving accuracy comparable to full precision networks across a range of popular models and datasets. We also show that exploiting these reduced-precision computational units in hardware can enable a super-linear improvement in inferencing performance due to a significant reduction in the area of accelerator compute engines coupled with the ability to retain the quantized model and activation data in on-chip memories.

## 1 Introduction

Deep Convolutional Neural Networks (CNNs) have achieved remarkable accuracy for tasks in a wide range of application domains including image processing (He et al. (2016b)), machine translation (Gehring et al. (2017)), and speech recognition (Zhang et al. (2017)). These state-of-the-art CNNs use very deep models, consuming 100s of ExaOps of computation during training and GBs of storage for model and data. This poses a tremendous challenge to widespread deployment, especially in resource constrained edge environments - leading to a plethora of explorations in compressed models that minimize memory footprint and computation while preserving model accuracy as much as possible.

Recently, a whole host of different techniques have been proposed to alleviate these computational costs. Among them, reducing the bit-precision of key CNN data structures, namely weights and activations, has gained attention due to its potential to significantly reduce both storage requirements and computational complexity. In particular, several weight quantization techniques (Li & Liu (2016) and Zhu et al. (2017)) showed significant reduction in the bit-precision of CNN weights with limited accuracy degradation. However, prior work (Hubara et al. (2016b); Zhou et al. (2016)) has shown that a straightforward extension of weight quantization schemes to activations incurs significant accuracy degradation in large-scale image classification tasks such as ImageNet (Russakovsky et al. (2015)). Recently, activation quantization schemes based on greedy layer-wise optimization were proposed (Park et al. (2017); Graham (2017); Cai et al. (2017)), but achieve limited accuracy improvement.

In this paper, we propose a novel activation quantization technique, PArameterized Clipping acTivation function (PACT), that automatically optimizes the quantization scales during model training. PACT allows significant reductions in the bit-widths needed to represent both weights and activations and opens up new opportunities for trading off hardware complexity with model accuracy.

The primary contributions of this work include:

1) PACT: A new activation quantization scheme for finding the optimal quantization scale during training. We introduce a new parameter $\alpha$ that is used to represent the clipping level

in the activation function and is learnt via back-propagation. $\alpha$ sets the quantization scale smaller than ReLU to reduce the quantization error, but larger than a conventional clipping activation function (used in previous schemes) to allow gradients to flow more effectively. In addition, regularization is applied to $\alpha$ in the loss function to enable faster convergence. We provide reasoning and analysis on the expected effectiveness of PACT in preserving model accuracy.

3) Quantitative results demonstrating the effectiveness of PACT on a spectrum of models and datasets. Empirically, we show that: (a) for extremely low bit-precision ($\leq$ 2-bits for weights and activations), PACT achieves the highest model accuracy compared to all published schemes and (b) 4-bit quantized CNNs based on PACT achieve accuracies similar to single-precision floating point representations.

4) System performance analysis to demonstrate the trade-offs in hardware complexity for different bit representations vs. model accuracy. We show that a dramatic reduction in the area of the computing engines is possible and use it to estimate the achievable system-level performance gains.

The rest of the paper is organized as follows: Section 2 provides a summary of related prior work on quantized CNNs. Challenges in activation quantization are presented in Section 3. We present PACT, our proposed solution for activation quantization in Section 4. In Section 5 we demonstrate the effectiveness of PACT relative to prior schemes using experimental results on popular CNNs. Overall system performance analysis for a representative hardware system is presented in Section 6 demonstrating the observed trade-offs in hardware complexity for different bit representations.

## 2 RELATED WORK

Recently, a whole host of different techniques have been proposed to minimize CNN computation and storage costs. One of the earliest studies in weight quantization schemes (Hwang & Sung (2014) and Courbariaux et al. (2015)) show that it is indeed possible to quantize weights to 1-bit (binary) or 2-bits (ternary), enabling an entire DNN model to fit effectively in resource-constrained platforms (e.g., mobile devices). Effectiveness of weight quantization techniques has been further improved (Li & Liu (2016) and Zhu et al. (2017)), by ternarizing weights using statistical distribution of weight values or by tuning quantization scales during training. However, gain in system performance is limited when only weights are quantized while activations are left in high precision. This is particularly severe in convolutional neural networks (CNNs) since weights are relatively smaller in convolution layers in comparison to fully-connected (FC) layers.

To reduce the overhead of activations, prior work (Kim & Smaragdis (2015), Hubara et al. (2016a), and Rastegari et al. (2016)) proposed the use of fully binarized neural networks where activations are quantized using 1-bit as well. More recently, activation quantization schemes using more general selections in bit-precision (Hubara et al. (2016b); Zhou et al. (2016; 2017); Mishra et al. (2017); Mellempudi et al. (2017)) have been studied. However, these techniques show significant degradation in accuracy ($> 1\%$) for ImageNet tasks (Russakovsky et al. (2015)) when bit precision is reduced significantly ($\leq 2 - bits$). Improvements to previous logarithmic quantization schemes (Miyashita et al. (2016)) using modified base and offset based on "weighted entropy" of activations have also been studied (Park et al. (2017)). Graham (2017) recommends that normalized activation, in the process of batch normalization (Ioffe & Szegedy (2015), BatchNorm), is a good candidate for quantization. Cai et al. (2017) further exploits the statistics of activations and proposes variants of the ReLU activation function for better quantization. However, such schemes typically rely on local (and greedy) optimizations, and are therefore not adaptable or optimized effectively during training. This is further elaborated in Section 3 where we present a detailed discussion on the challenges in quantizing activations.

## 3 CHALLENGES IN ACTIVATION QUANTIZATION

Quantization of weights is equivalent to discretizing the hypothesis space of the loss function with respect to the weight variables. Therefore, it is indeed possible to compensate weight quantization errors during model training (Hwang & Sung, 2014; Courbariaux et al., 2015). Traditional activation

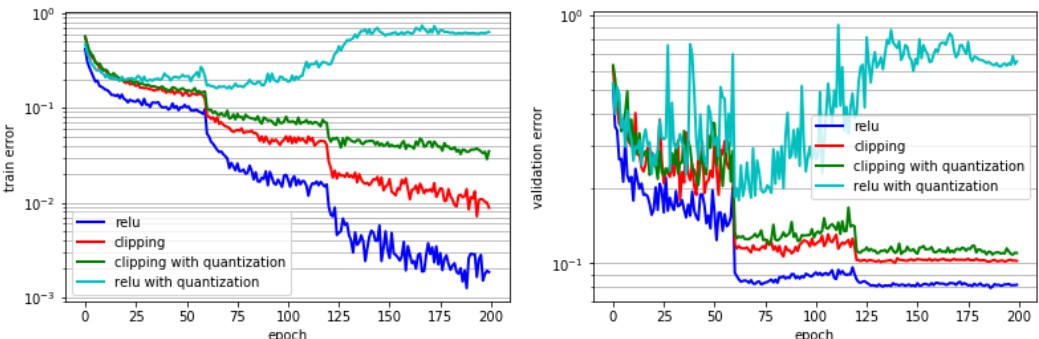

Figure 1: (a) Training error, (b) Validation error across epochs for different activation functions (relu and clipping) with and without quantization for the ResNet20 model using the CIFAR10 dataset

functions, on the other hand, do not have any trainable parameters, and therefore the errors arising from quantizing activations cannot be directly compensated using back-propagation.

Activation quantization becomes even more challenging when ReLU (the activation function most commonly used in CNNs) is used as the layer activation function (ActFn). ReLU allows gradient of activations to propagate through deep layers and therefore achieves superior accuracy relative to other activation functions (Nair & Hinton (2010)). However, as the output of the ReLU function is unbounded, the quantization after ReLU requires a high dynamic range (i.e., more bit-precision). In Fig. 1 we present the training and validation errors of ResNet20 with the CIFAR10 dataset using ReLU and show that accuracy is significantly degraded with ReLU quantizations

It has been shown that this dynamic range problem can be alleviated by using a clipping activation function, which places an upper-bound on the output (Hubara et al. (2016b); Zhou et al. (2016)). However, because of layer to layer and model to model differences - it is difficult to determine a globally optimal clipping value. In addition, as shown in Fig 1, even though the training error obtained using clipping with quantization is less than that obtained with quantized ReLU, the validation error is still noticeably higher than the baseline.

Recently, this challenge has been partially addressed by applying a half-wave Gaussian quantization scheme to activations (Cai et al. (2017)). Based on the observation that activation after BatchNorm normalization is close to a Gaussian distribution with zero mean and unit variance, they used Lloyd's algorithm to find the optimal quantization scale for this Gaussian distribution and use that scale for every layer. However, this technique also does not fully utilize the strength of backpropagation to optimally learn the clipping level because all the quantization parameters are determined offline and remain fixed throughout the training process.

## 4 PACT: Parameterized Clipping Activation Function

Building on these insights, we introduce PACT, a new activation quantization scheme in which the ActFn has a parameterized clipping level, $\alpha$. $\alpha$ is dynamically adjusted via gradient descent-based training with the objective of minimizing the accuracy degradation arising from quantization. In PACT, the conventional ReLU activation function in CNNs is replaced with the following:

$$y = PACT(x) = 0.5(|x| - |x - \alpha| + \alpha) = \begin{cases} 0, & x \in (-\infty, 0) \\ x, & x \in [0, \alpha) \\ \alpha, & x \in [\alpha, +\infty) \end{cases} \quad (1)$$

where $\alpha$ limits the range of activation to $[0, \alpha]$. The truncated activation output is then linearly quantized to $k$ bits for the dot-product computations, where

$$y_q = round(y \cdot \frac{2^k - 1}{\alpha}) \cdot \frac{\alpha}{2^k - 1} \qquad (2)$$

With this new activation function, $\alpha$ is a variable in the loss function, whose value can be optimized during training. For back-propagation, gradient $\frac{\partial y_q}{\partial \alpha}$ can be computed using the Straight-Through Estimator (STE) (Bengio et al. (2013)) to estimate $\frac{\partial y_q}{\partial y}$ as 1. Thus,

$$\frac{\partial y_q}{\partial \alpha} = \frac{\partial y_q}{\partial y} \frac{\partial y}{\partial \alpha} = \begin{cases} 0, & x \in (-\infty, \alpha) \\ 1, & x \in [\alpha, +\infty) \end{cases} \qquad (3)$$

The larger the $\alpha$, the more the parameterized clipping function resembles a ReLU Actfn. To avoid large quantization errors due to a wide dynamic range, we include a L2-regularizer for $\alpha$ in the loss function. Fig. 7 illustrates how the value of $\alpha$ changes during full-precision training of CIFAR10-ResNet20 starting with an initial value of 10 and using the L2-regularizer. It can be observed that $\alpha$ converges to values much smaller than the initial value as the training epochs proceed, thereby limiting the dynamic range of activations and minimizing quantization loss.

To provide further reasoning on why PACT works, we provide in-depth analysis in Appendix A and B. In particular, we show in Appendix A that PACT is as expressive as ReLU when it is used as an activation function. Further we explain in Appendix B that PACT finds a balancing point between clipping and quantization errors to minimize their impact to classification accuracy.

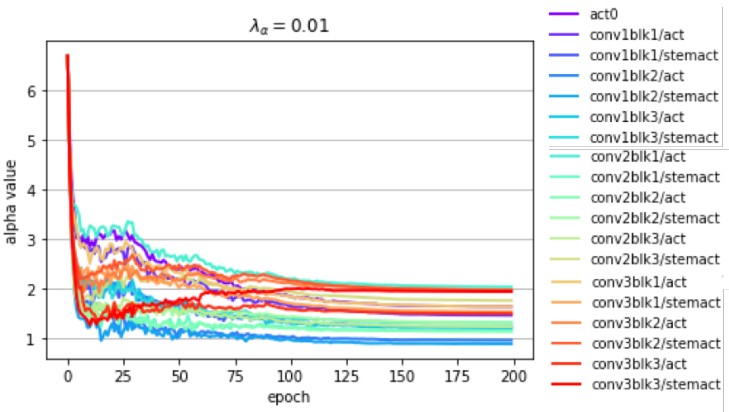

Figure 2: Evolution of $\alpha$ values during training using a ResNet20 model on the CIFAR10 dataset.

## 4.1 UNDERSTANDING HOW PARAMETERIZED CLIPPING WORKS

When activation is quantized, the overall behavior of network parameters is affected by the quantization error during training. To observe the impact of activation quantization during network training, we sweep the clipping parameter $\alpha$ and record the training loss with and without quantization. Figs. 3 a,b and 3c show cross-entropy and training loss (cross entropy + regularization), respectively, over a range of $\alpha$ for the pre-trained SVHN network. The loaded network is trained with the proposed quantization scheme in which ReLU is replaced with the proposed parameterized clipping ActFn for each of its seven convolution layers. We sweep the value of $\alpha$ one layer at a time, keeping all other parameters (weight ($W$), bias ($b$), BatchNorm parameters ($\beta, \gamma$), and the $\alpha$ of other layers) fixed when computing the cross-entropy and training loss.

The cross-entropy computed via full-precision forward-pass of training is shown in Fig. 7b. In this case, the cross-entropy converges to a small value in many layers as $\alpha$ increases, indicating that ReLU is a good activation function when no quantization is applied. But even for the full-precision

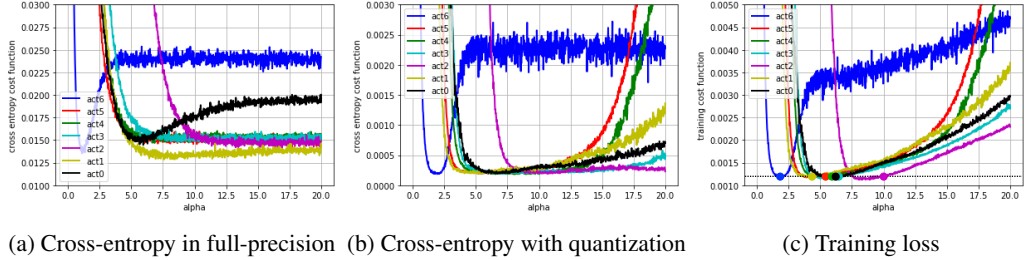

(a) Cross-entropy in full-precision (b) Cross-entropy with quantization (c) Training loss

Figure 3: Cross-entropy vs $\alpha$ for SVHN image classification.

case, training clipping parameter $\alpha$ may help reduce the cross-entropy for certain layers; for example, ReLU (i.e., $\alpha = \infty$) is not optimal for act0 and act6 layers.

Next, the cross-entropy computed with quantization in the forward-pass is shown in Fig. 3b. With quantization, the cross-entropy increases in most cases as $\alpha$ increases, implying that ReLU is no longer effective. We also observe that the optimal $\alpha$ has different ranges for different layers, motivating the need to "learn" the quantization scale via training. In addition, we observe plateaus of cross-entropy for the certain ranges of $\alpha$ (e.g., act6), leading to difficulties for gradient descent-based training.

Finally, in Fig. 3c, we show the total training loss including both the cross-entropy discussed above and the cost from $\alpha$ regularization. The regularization effectively gets rid of the plateaus in the training loss, thereby favoring convergence for gradient-descent based training. At the same time, $\alpha$ regularization does not perturb the global minimum point. For example, the solid circles in Fig. 3c, which are the optimal $\alpha$ extracted from the pre-trained model, are at the minimum of the training loss curves. The regularization coefficient, $\lambda_\alpha$, discussed in the next section, is an additional hyper-parameter which controls the impact of regularization on $\alpha$.

### 4.2 EXPLORATION OF HYPER-PARAMETERS

For this new quantization approach, we studied the scope of $\alpha$, the choice of initial values of $\alpha$, and the impact of regularizing $\alpha$. We briefly summarize our findings below, and present more detailed analysis in Appendix C.

From our experiments, the best scope for $\alpha$ was to share $\alpha$ per layer. This choice also reduces hardware complexity because $\alpha$ needs to be multiplied only once after all multiply-accumulate (MAC) operations in reduced-precision in a layer are completed.

Among initialization choices for $\alpha$, we found it to be advantageous to initialize $\alpha$ to a larger value relative to typical values of activation, and then apply regularization to reduce it during training.

Finally, we observed that applying L2-regularization for $\alpha$ with the same regularization parameter $\lambda$ used for weight works reasonably well. We also observed that, as expected, the optimal value for $\lambda_\alpha$ slightly decreases when higher bit-precision is used because more quantization levels result in higher resolution for activation quantization.

Additionally, we follow the practice of many other quantized CNN studies (e.g., Hubara et al. (2016b); Zhou et al. (2016)), and do not quantize the first and last layers, as these have been reported to significantly impact accuracy.

### 5 EXPERIMENTS

We implemented PACT in Tensorflow (Abadi et al. (2015)) using Tensorpack (Zhou et al. (2016)). To demonstrate the effectiveness of PACT, we studied several well-known CNNs. The following is a summary of the Dataset-Network for the tested CNNs. More implementation details can be found in Appendix.D. Note that the baseline networks use the same hyper-parameters and ReLU activation functions as described in the references. For PACT experiments, we only replace ReLU into PACT but the same hyper-parameters are used. All the time the networks are trained from scratch.

- CIFAR10-ResNet20 (CIFAR10, Krizhevsky & Hinton (2010)): a convolution (CONV) layer followed by 3 ResNet blocks (16 CONV layers with 3x3 filter) and a final fully-connected (FC) layer.

- SVHN-SVHN (SVHN, Netzer et al. (2011)): 7 CONV layers followed by 1 FC layer.

- IMAGENET-AlexNet (AlexNet, Krizhevsky et al. (2012)): 5 parallel-CONV layers followed by 3 FC layers. BatchNorm is used before ReLU.

- IMAGENET-ResNet18 (ResNet18, He et al. (2016b)): a CONV layer followed by 8 ResNet blocks (16 CONV layers with 3x3 filter) and a final FC layer. "full pre-activation" ResNet structure (He et al. (2016a)) is employed.

- IMAGENET-ResNet50 (ResNet50, He et al. (2016b)): a CONV layer followed by 16 ResNet "bottleneck" blocks (total 48 CONV layers) and a final FC layer. "full pre-activation" ResNet structure (He et al. (2016a)) is employed.

For comparisons, we include accuracy results reported in the following prior work: DoReFa (Zhou et al. (2016)), BalancedQ (Zhou et al. (2017)), WRPN (Mishra et al. (2017)), FGQ (Mellempudi et al. (2017)), WEP (Park et al. (2017)), LPBN (Graham (2017)), and HWGQ (Cai et al. (2017)). Detailed experimental setting for each of these papers, as well as full comparison of accuracy (top-1 and top5) for AlexNet, ResNet18, ResNet50, can be found in Appendix E. In the following section, we present key results demonstrating the effectiveness of PACT relative to prior work.

## 5.1 Activation Quantization Performance

We first evaluate our activation quantization scheme using various CNNs. Fig 4 shows training and validation error of PACT for the tested CNNs. Overall, the higher the bit-precision, the closer the training/validation errors are to the full-precision reference. Specifically it can be seen that training using bit-precision higher than 3-bits converges almost identically to the full-precision baseline. The final validation error has less than $1\%$ difference relative to the full-precision validation error for all cases when the activation bit-precision is at least 4-bits.

We further compare activation quantization performance with 3 previous schemes, DoReFa, LPBN, and HWGQ. We use *accuracy degradation* as the quantization performance metric, which is calculated as the difference between full-precision accuracy and the accuracy for each quantization bit-precision. Fig. 4f shows accuracy degradation (top-1) for ResNet18 (left) and ResNet50 (right) for increasing activation bit-precision, when the same weight bit-precision is used for each quantization scheme (indicated within the parenthesis). Overall, we observe that accuracy degradation is reduced as we increase the bit-precision of activations. For both ResNet18 and ResNet50, PACT achieves consistently lower accuracy degradation compared to the other quantization schemes, demonstrating the robustness of PACT relative to prior quantization approaches.

## 5.2 PACT Performance for Quantized CNNs

In this section, we demonstrate that although PACT targets activation quantization, it does not preclude us from using weight quantization as well. We used PACT to quantize activation of CNNs, and DoReFa scheme to quantize weights. Table 1 summarizes top-1 accuracy of PACT for the tested CNNs (CIFAR10, SVHN, AlexNet, ResNet18, and ResNet50). We also show the accuracy of CNNs when both the weight and activation are quantized by DoReFa's scheme. As can be seen, with 4 bit precision for both weights and activation, PACT achieves full-precision accuracy consistently across the networks tested. To the best of our knowledge, this is the lowest bit precision for both weights and activation ever reported, that can achieve near ($\leq 1\%$) full-precision accuracy.

We further compare the performance of PACT-based quantized CNNs with 7 previous quantization schemes (DoReFa, BalancedQ, WRPN, FGQ, WEP, LPBN, and HWGQ). Fig. 5 shows comparison of accuracy degradation (top-1) for AlexNet, ResNet18, and ResNet50. Overall, the accuracy degradation decreases as bit-precision for activation or weight increases. For example, in Fig. 5a, the accuracy degradation decreases when activation bit-precision increases given the same weight precision or when weight bit-precision increases given the same activation bit-precision. PACT outperforms other schemes for all the cases. In fact, AlexNet even achieves marginally better accuracy (i.e., negative accuracy degradation) using PACT instead of full-precision.

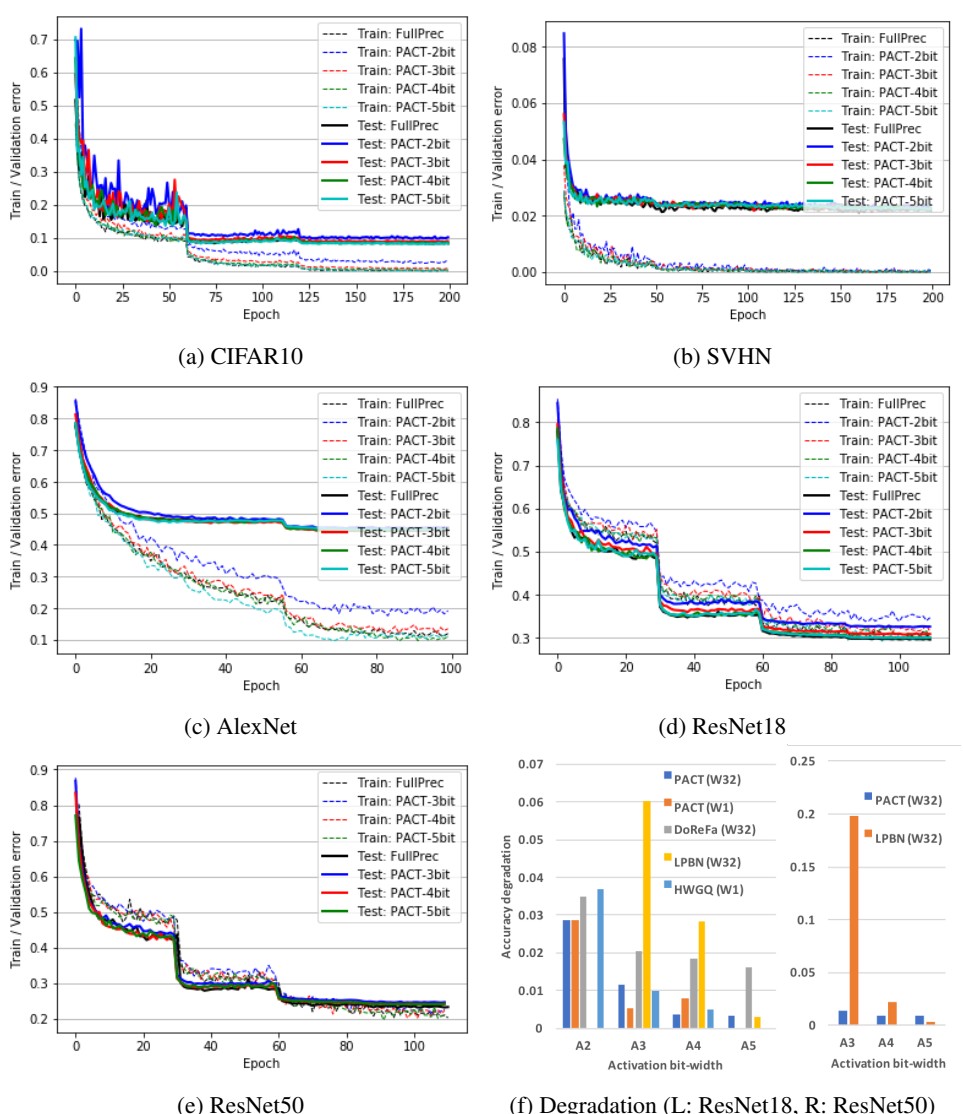

Figure 4: (a-e) Training and valid error with different bit-precision for various CNNs. (f) Comparison of accuracy degradation for ResNet18 (left) and ResNet50 (right). The lower the better.

# 6 SYSTEM-LEVEL PERFORMANCE GAIN

In this section, we demonstrate the gain in system performance as a result of the reduction in bit-precision achieved using PACT-CNN. To this end, as shown in Fig. 6(a), we consider a DNN accelerator system comprising of a DNN accelerator chip, comprising of multiple cores, interfaced with an external memory. Each core consists of a 2D-systolic array of fixed-point multiply-and-accumulate (MAC) processing elements on which DNN layers are executed. Each core also contains an on-chip memory, which stores the operands that are fed into the MAC processing array.

To estimate system performance at different bit precisions, we studied different versions of the DNN accelerator each comprising the same amount of on-chip memory, external memory bandwidth, and occupying iso-silicon area. First, using real hardware implementations in a state of the art technology (14 nm CMOS), we accurately estimate the reduction in the MAC area achieved by aggressively scaling bit precision. As shown in Fig. 6(b), we achieve ∼14× improvement in density when the bit-precisions of both activations and weights are uniformly reduced from 16 bits to 2 bits.

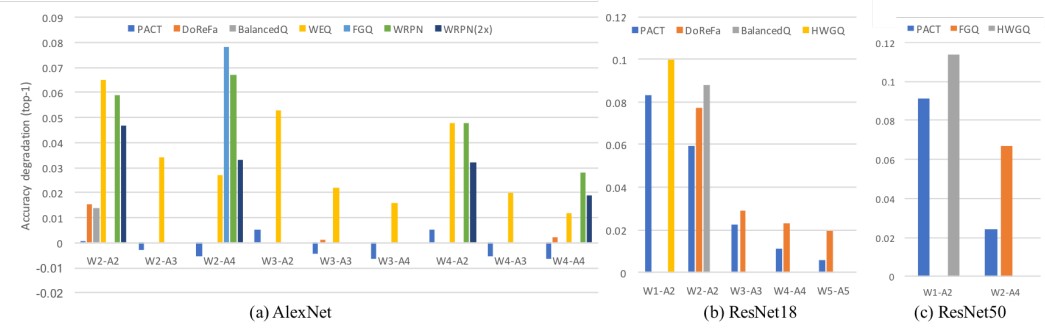

Figure 5: Comparison of accuracy degradation (Top-1) for (a) AlexNet, (b) ResNet18, and (c) ResNet50.

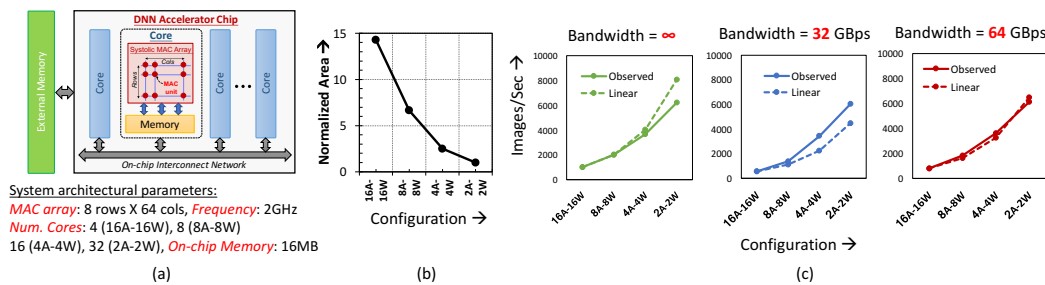

Figure 6: (a)System architecture and parameters, (b) Variation in MAC area with bit-precision and (b) Speedup at different quantizations for inference using ResNet50 DNN

Next, to translate the reduction in area to improvement in overall performance, we built a precision-configurable MAC unit, whose bit precision can be modulated dynamically. The peak compute capability (FLOPs) of the MAC unit varied such that we achieve iso-area at each precision. Note that the total on-chip memory and external bandwidth remains constant at all precisions. We estimate the overall system performance using DeepMatrix, a detailed performance modelling framework for DNN accelerators (Venkataramani et al.).

Fig. 6(c) shows the gain in inference performance for the ResNet50 DNN benchmark. We study the performance improvement using different external memory bandwidths, namely, a bandwidth unconstrained system (infinite memory bandwidth) and two bandwidth constrained systems at 32 and 64 GBps. In the bandwidth unconstrained scenario, the gain in performance is limited by how amenable it is to parallelize the work. In this case, we see a near-linear increase in performance for upto 4 bits and a small drop at extreme quantization levels (2 bits).

Practical systems, whose bandwidths are constrained, (surprisingly) exhibit a super-linear growth in performance with quantization. For example, when external bandwidth is limited to 64 GBps, quantizing from 16 to 4 bits leads to a $4\times$ increase in peak FLOPs but a $4.5\times$ improvement in performance. This is because, the total amount of on-chip memory remains constant, and at very low precision some of the data-structures begin to fit within the memory present in the cores, thereby *avoiding* data transfers from the external memory. Consequently, in bandwidth limited systems, reducing the amount of data transferred from off-chip can provide an additional boost in system performance beyond the increase in peak FLOPs. Note that for the 4 and 2 bit precision configurations, we still used 8 bit precision to execute the first and last layers of the DNN. If we are able to quantize the first and last layers as well to 4 or 2 bits, we estimate an additional $1.24\times$ improvement in performance, motivating the need to explore ways to quantize the first and last layers.

Table 1: Comparison of top-1 accuracy between DoReFa and PACT. Weights are quantized with DoReFa scheme, whereas activations are quantized with our scheme. Note that CNNs with $4b$ quantization based on our scheme achieves full-precision accuracy for all the CNNs we explored.

| Network | FullPrec | DoReFa | | | | PACT | | | |
|---|---|---|---|---|---|---|---|---|---|
| | | 2b | 3b | 4b | 5b | 2b | 3b | 4b | 5b |
| CIFAR10 | 0.916 | 0.882 | 0.899 | 0.905 | 0.904 | 0.897 | 0.911 | 0.913 | 0.917 |
| SVHN | 0.978 | 0.976 | 0.976 | 0.975 | 0.975 | 0.977 | 0.978 | 0.978 | 0.979 |
| AlexNet | 0.551 | 0.536 | 0.550 | 0.549 | 0.549 | 0.550 | 0.556 | 0.557 | 0.557 |
| ResNet18 | 0.702 | 0.626 | 0.675 | 0.681 | 0.684 | 0.644 | 0.681 | 0.692 | 0.698 |
| ResNet50 | 0.769 | 0.671 | 0.699 | 0.714 | 0.714 | 0.722 | 0.753 | 0.765 | 0.767 |

## 7 CONCLUSION

In this paper, we propose a novel activation quantization scheme based on the PArameterized Clipping acTivation function (PACT). The proposed scheme replaces ReLU with an activation function with a clipping parameter, $\alpha$, that is optimized via gradient descent based training. We provide analysis on why PACT outperforms ReLU when quantization is applied during training. Extensive empirical evaluation using several popular convolutional neural networks, such as CIFAR10, SVHN, AlexNet, ResNet18 and ResNet50, shows that PACT quantizes activations very effectively while simultaneously allowing weights to be heavily quantized. In comparison to all previous quantization schemes, we show that both weights and activations can be quantized much more aggressively (down to 4-bits) - while achieving near ($\leq 1\%$) full-precision accuracy. In addition, we have shown that the area savings from using reduced-precision MAC units enable a dramatic increase in the number of accelerator cores in the same area, thereby, significantly improving overall system-performance.

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

## APPENDIX A    PACT IS AS EXPRESSIVE AS ReLU

When used as an activation function of the neural network, PACT is as expressive as ReLU. This is because clipping parameter, $\alpha$, introduced in PACT, allows flexibility in adjusting the dynamic range of activation for each layer. We demonstrate in the simple example below that PACT can reach the same solution as ReLU via SGD.

**Lemma A.1.** *Consider a single-neuron network with PACT; $x = w \cdot a$, $y = PACT(x)$, where $a$ is input and $w$ is weight. This network can be trained with SGD to find the output the network with ReLU would produce.*

*Proof.* Consider a sample of training data $(a, y^*)$. For illustration purposes consider mean-square-error (MSE) as the cost function: $L = 0.5 \cdot (y^* - y)^2$.

If $x \leq \alpha$, then clearly the network with PACT behaves the same as the network with ReLU.

If $x > \alpha$, then $y = \alpha$ and $\frac{\partial y}{\partial \alpha} = 1$ from (1). Thus,

$$\frac{\partial L}{\partial \alpha} = \frac{\partial L}{\partial y} \cdot \frac{\partial y}{\partial \alpha} = \frac{\partial L}{\partial y} \tag{4}$$

Therefore, when $\alpha$ is updated by SGD,

$$\alpha_{new} = \alpha - \eta \frac{\partial L}{\partial \alpha} = \alpha - \eta \frac{\partial L}{\partial y} \tag{5}$$

where $\eta$ is a learning rate. Note that during this update, the weight is not updated as $\frac{\partial L}{\partial w} = \frac{\partial L}{\partial y} \cdot \frac{\partial y}{\partial x} (= 0) \cdot a = 0$.

From MSE, $\frac{\partial L}{\partial y} = (y - y^*)$. Therefore, if $y^* > x$, $\alpha$ is increased for each update of (5) until $\alpha \geq x$, then the PACT network behaves the same as the ReLU network.

Interestingly, if $y^* \leq y$ or $y < y^* < x$, $\alpha$ is decreased or increased to converge to $y^*$. Note that in this case, ReLU would pass erroneous output $x$ to increase cost function, which needs to be fixed by updating $w$ with $\frac{\partial L}{\partial w}$. PACT, on the other hand, ignores this erroneous output by directly adapting the dynamic range to match the target output $y^*$. In this way, the PACT network can be trained to produce output which converges to the same target that the ReLU network would achieve via SGD.

$\square$

In general cases, $\frac{\partial L}{\partial \alpha} = \sum_i \frac{\partial L}{\partial y_i}$, and PACT considers output of neurons together to change the dynamic range. There are two options: (1) if output $x_i$ is not clipped, then the network is trained via back-propagation of gradient to update weight, (2) if output $x_i$ is clipped, then $\alpha$ is increased or decreased based on how close the overall output is to the target. Hence, there are configurations under which SGD would lead to a solution close to the one which the network with ReLU would achieve. Fig. 7 demonstrates that ResNet20 with PACT converges almost identical to the network with ReLU.

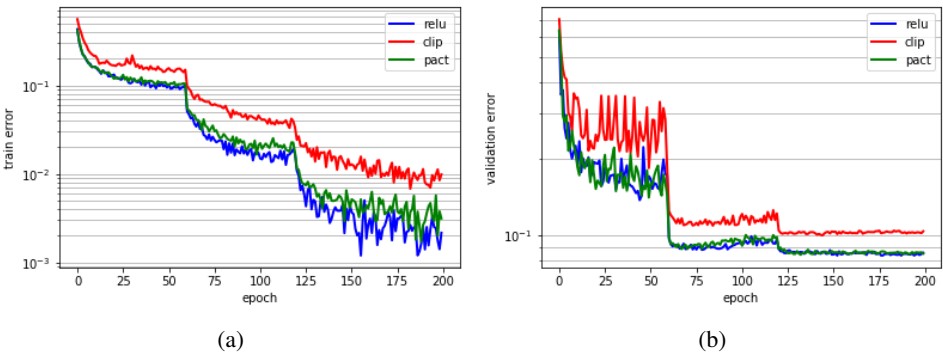

(a)                      (b)

Figure 7: (a) Training error and (b) validation error of PACT for ResNet20 model on the CIFAR10 dataset. Note that the convergence curve for PACT is almost identical to ReLU, although the dynamic range via trained clipping levels are much lower than ReLU.

## APPENDIX B    PACT FOR BALANCING CLIPPING AND QUANTIZATION ERRORS

In Section 3, when we briefly discussed the challenges in activation quantization, we mentioned that there is a trade-off between errors due to clipping and quantization. As the clipping level increases, larger range of activation can be passed to the next layer of the neural network causing less clipping error ($ErrClip_i = max(x_i - \alpha, 0)$). However, the increased dynamic range incurs larger quantization error, since its magnitude is proportional to the clipping level ($ErrQuant_i \leq 0.5 \cdot \frac{\alpha}{2^k - 1}$, with $k$-bit quantization). This imposes the challenge of finding a proper clipping level to balance between clipping and quantization errors.

This trade-off can be better observed in Fig. 8a, which shows normalized mean-square-error caused by clipping and quantization during training of the CIFAR10-ResNet20 with different clipping levels. It can be seen that activation functions with large dynamic range, such as ReLU, would suffer quantization errors whose magnitude increases exponentially as the bit-precision $k$ decreases. This explains why the network with ReLU fails to converge when the activation is quantized (Fig. 1).

PACT can find a balancing point between clipping and quantization errors. As explained in Section A, PACT adjusts dynamic range based on how close the output is to the target. As both clipping and quantization errors distort output far from the target, PACT would increase or decrease the dynamic range during training to minimize both clipping and quantization errors.

Fig. 8b shows how PACT balances the clipping and quantization errors during training. CIFAR10-ResNet20 is trained with clipping activation function with varying clipping level $\alpha$ from 1 to 16. When activation is quantized, the network trained with clipping activation shows significant accuracy degradation as $\alpha$ increases. This is consistent with the trend in quantization error we observed in Fig. 8a. In this case, PACT achieves the best accuracy one of the clipping activation could achieve, but without exhaustively sweeping over different clipping levels. In other words, PACT auto-tunes the clipping level to achieve best accuracy without incurring significant computation overhead. PACT's auto-tuning of dynamic range is critical in efficient yet robust training of large scale quantized neural networks, especially because it does not increase the burden for hyper-parameter tuning. In fact, we used the same hyper-parameters as well as the original network structure for all the models we tested, except replacing ReLU to PACT, when we applied activation quantization.

Without quantization, there is a trend that validation error decreases as $\alpha$ increases. Surprisingly, some of the cases even outperforms the ReLU network. In this case, PACT also achieves comparable accuracy as ReLU, confirming its expressivity discussed in Section A.

## APPENDIX C    EXPLORATION OF HYPER-PARAMETERS AND DESIGN CHOICES

In this section, we present details on the hyper-parameters and design choices studied for PACT.

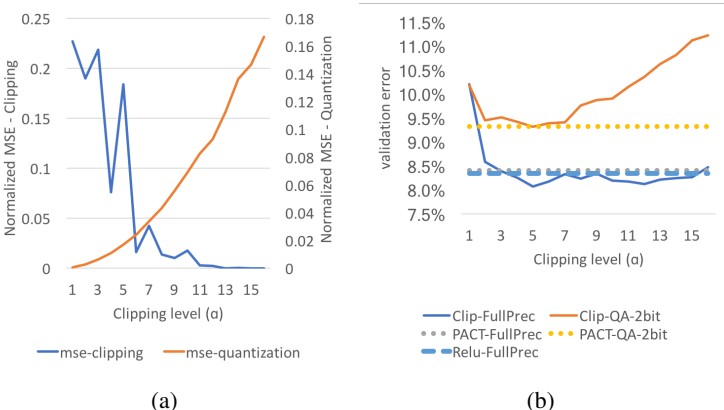

(a)                                        (b)

Figure 8: Experiment on CIFAR10-ResNet20 to validate that PACT balances clipping and quantization errors. (a) Trade-off between clipping and quantization error. (b) PACT achieving lowest validation error that clipping activation can achieve without exhaustive search over clipping level $\alpha$.

## C.1   SCOPE OF $\alpha$

One of key questions is the optimal scope for $\alpha$. In other words, determining which neuron activations should share the same $\alpha$. We considered 3 possible choices: (a) Individual $\alpha$ for each neuron activation, (b) Shared $\alpha$ among neurons within the same output channel, and (c) Shared $\alpha$ within a layer. We empirically studied each of these choices of $\alpha$ (without quantization) using CIFAR10-ResNet20 and determined training and validation error for PACT. As shown in Fig. 9, sharing $\alpha$ per layer is the best choice in terms of accuracy. This is in fact a preferred option from the perspective of hardware complexity as well, since $\alpha$ needs to be multiplied only once after all multiply-accumulate(MAC) operations in a layer are completed.

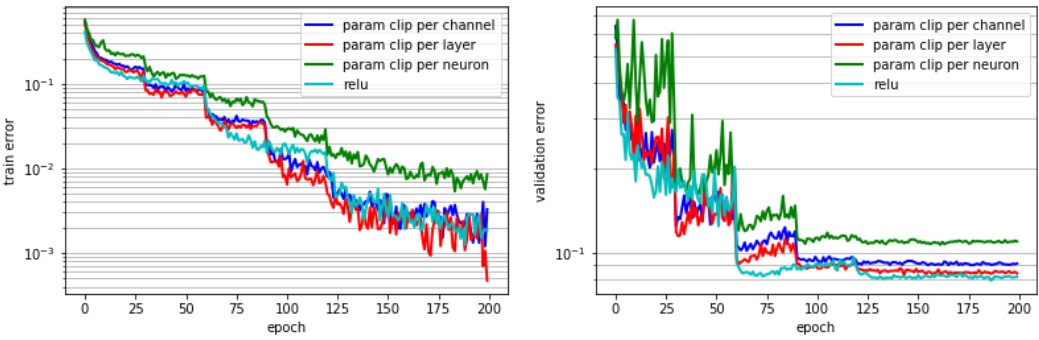

Figure 9: Training and validation error of CIFAR10-ResNet20 for PACT with different scope of $\alpha$.

## C.2   INITIAL VALUE AND REGULARIZATION OF $\alpha$

The optimization behavior of $\alpha$ can be explained from the formulation of the parameterized clipping function. From Eq. 3 it is clear that, if $\alpha$ is initialized to a very small value, more activations fall into the range for the nonzero gradient, leading to unstable $\alpha$ in the early epochs, potentially causing accuracy degradation. On the other hand, if $\alpha$ is initialized to a very large value, the gradient becomes too small and $\alpha$ may be stuck at a large value, potentially suffering more on quantization error. Therefore, it is intuitive to start with a reasonably large value to cover a wide dynamic range and avoid unstable adaptation of $\alpha$, but apply regularizer to reduce the value of $\alpha$ so as to alleviate quantization error.

In practice, we found that applying L2-regularization for $\alpha$ while setting its coefficient $\lambda_\alpha$ the same as the L2-regularization coefficient for weight, $\lambda$, works well. Fig. 10 shows that validation error for

PACT-quantized CIFAR10-ResNet20 does not significantly vary for a wide range of $\lambda_\alpha$. We also observed that, as expected, the optimal value for $\lambda_\alpha$ slightly decreases when higher bit-precision is used because more quantization levels result in higher resolution for activation quantization.

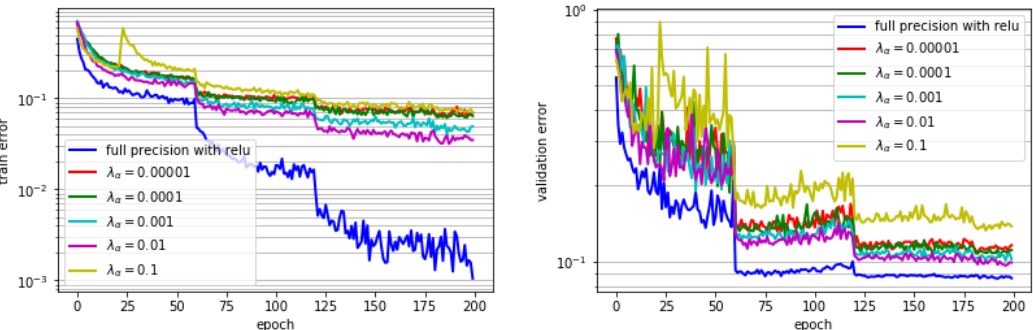

Figure 10: Training and validation error of quantized CIFAR10-ResNet20 for PACT with different regularization parameter $\lambda_\alpha$.

### C.3 QUANTIZATION OF FIRST AND LAST LAYERS

Many previous work (e.g., Hubara et al. (2016b); Zhou et al. (2016)) follow the convention to keep the first and last layer in full precision during training, since quantizing those layers lead to substantial accuracy degradation. We empirically studied this for the proposed quantization approach for CIFAR10-ResNet20. In Fig. 11, the only difference among the curves is whether input activation and weight of the first convolution layer or the last fully-connected layer are quantized. As can be seen from the plots, there can be noticeable accuracy degradation if the first or last layers are aggressively quantized. But computation in floating point is very expensive in hardware.

Therefore, we further studied the option of quantizing the first and last layers with higher quantization bit-precision than the bit-precision of the other layers. Table 2 shows that independent of the quantization level for the other layers, there is little accuracy degradation if the first and last layer are quantized with 8-bits. This motivates us to employ reduced precision computation even for the first/last layers.

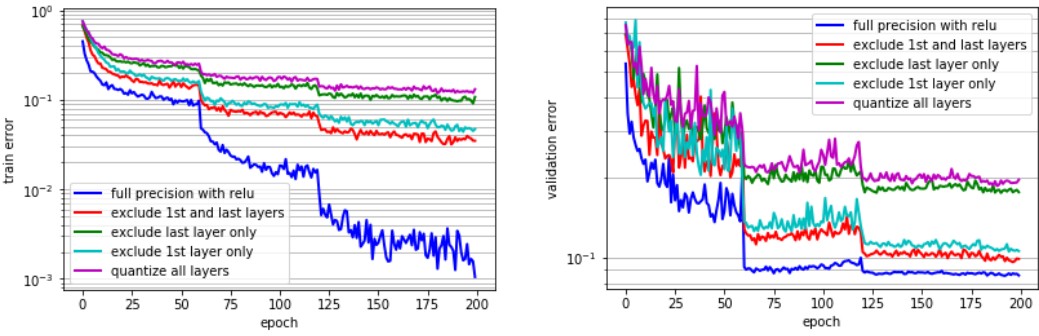

Figure 11: Comparison of accuracy of CIFAR10-ResNet20 with and without quantization of the first and last layers.

## APPENDIX D  CNN IMPLEMENTATION DETAILS

In this section, we summarize details of our CNN implementation as well as our training settings, which is based on the default networks provided by Tensorpack (Zhou et al. (2016)). Unless mentioned otherwise, ReLU following BatchNorm is used for ActFn of the convolution (CONV)

Table 2: Validation error (in %) of CIFAR10-ResNet20 when first and last layers are quantized with different bit-precision. FL/M/FL means the first and last layers are quantized with Bit-FL bits, while the other layers are quantized with Bit-M bits. NQ represents no quantization is applied.

| BIT-M (bits) | 2 | | | | 3 | | | | 4 | | | | 5 | | | |
|---|---|---|---|---|---|---|---|---|---|---|---|---|---|---|---|---|
| BIT-FL (bits) | 2 | 4 | 8 | 32 | 2 | 4 | 8 | 32 | 2 | 4 | 8 | 32 | 2 | 4 | 8 | 32 |
| FL/M/FL | 21.0 | 12.9 | 11.1 | 10.9 | 17.4 | 10.0 | 9.4 | 8.9 | 15.9 | 9.7 | 9.2 | 8.9 | 18.2 | 9.0 | 8.4 | 8.5 |
| FL/M/NQ | 21.3 | 11.5 | 11.5 | 10.7 | 17.6 | 9.7 | 9.2 | 9.0 | 16.5 | 9.7 | 8.7 | 8.7 | 16.3 | 9.3 | 8.6 | 8.5 |
| NQ/M/FL | 12.1 | 11.2 | 11.0 | 11.5 | 9.8 | 8.9 | 9.2 | 9.2 | 8.4 | 8.4 | 8.7 | 8.8 | 8.5 | 9.0 | 8.5 | 8.5 |

layers, and Softmax is used for the fully-connected (FC) layer. Note that the baseline networks use the same hyper-parameters and ReLU activation functions as described in the references. For PACT experiments, we only replace ReLU into PACT but the same hyper-parameters are used. All the time the networks are trained from scratch.

The CIFAR10 dataset (Krizhevsky & Hinton (2010)) is an image classification benchmark containing $32 \times 32$ pixel RGB images. It consists of 50K training and 10K test image sets. We used the "standard" ResNet structure (He et al. (2016a)) which consists of a CONV layer followed by 3 ResNet blocks (16 CONV layers with 3x3 filter) and a final FC layer. We used stochastic gradient descent (SGD) with momentum of 0.9 and learning rate starting from 0.1 and scaled by 0.1 at epoch 60, 120. L2-regularizer with decay of 0.0002 is applied to weight. The mini-batch size of 128 is used, and the maximum number of epochs is 200.

The SVHN dataset (Netzer et al. (2011)) is a real-world digit recognition dataset containing photos of house numbers in Google Street View images, where the "cropped" $32 \times 32$ colored images (resized to $40 \times 40$ as input to the network) centered around a single character are used. It consists of 73257 digits for training and 26032 digits for testing. We used a CNN model which contains 7 CONV layers followed by 1 FC layer. We used ADAM(Kingma & Ba (2015)) with epsilon $10^{-5}$ and learning rate starting from $10^{-3}$ and scaled by 0.5 every 50 epoch. L2-regularizer with decay of $10^{-7}$ is applied to weight. The mini-batch size of 128 is used, and the maximum number of epochs is 200.

The IMAGENET dataset (Russakovsky et al. (2015)) consists of 1000-categories of objects with over 1.2M training and 50K validation images. Images are first resized to 256 256 and randomly cropped to 224224 prior to being used as input to the network. We used a modified AlexNet, ResNet18 and ResNet50.

We used AlexNet network (Krizhevsky et al. (2012)) in which local contrast renormalization (R-Norm) layer is replaced with BatchNorm layer. We used ADAM with epsilon $10^{-5}$ and learning rate starting from $10^{-4}$ and scaled by 0.2 at epoch 56 and 64. L2-regularizer with decay factor of $5 \times 10^{-6}$ is applied to weight. The mini-batch size of 128 is used, and the maximum number of epochs is 100.

ResNet18 consists of a CONV layer followed by 8 ResNet blocks (16 CONV layers with 3x3 filter) and a final FC layer. "full pre-activation" ResNet structure (He et al. (2016a)) is employed. ResNet50 consists of a CONV layer followed by 16 ResNet "bottleneck" blocks (total 48 CONV layers) and a final FC layer. "full pre-activation" ResNet structure (He et al. (2016a)) is employed.

For both ResNet18 and ResNet50, we used stochastic gradient descent (SGD) with momentum of 0.9 and learning rate starting from 0.1 and scaled by 0.1 at epoch 30, 60, 85, 95. L2-regularizer with decay of $10^{-4}$ is applied to weight. The mini-batch size of 256 is used, and the maximum number of epochs is 110.

## APPENDIX E   COMPARISON WITH RELATED WORK

### E.1   QUANTIZATION EXPERIMENT SETTING

- DoReFa-Net (DoReFa, Zhou et al. (2016)): A general bit-precision uniform quantization schemes for weight, activation, and gradient of DNN training.We compared the experimental results of DoReFa for CIFAR10, SVHN, AlexNet and ResNet18 under the same experimental

setting as PACT. Note that a clipped absolute activation function is used for SVHN in DoReFa.

- Balanced Quantization (BalancedQ, Zhou et al. (2017)): A quantization scheme based on recursive partitioning of data into balanced bins. We compared the reported top-1/top-5 validation accuracy of their quantization scheme for AlexNet and ResNet18.

- Quantization using Wide Reduced-Precision Networks (WRPN, Mishra et al. (2017)): A scheme to increase the number of filter maps to increase robustness for activation quantization. We compared the reported top-1 accuracy of their quantization with various weight/activation bit-precision for AlexNet.

- Fine-grained Quantization (FGQ, Mellempudi et al. (2017)): A direct quantization scheme (i.e., little re-training needed) based on fine-grained grouping (i.e., within a small subset of filter maps). We compared the reported top-1 validation accuracy of their quantization with 2-bit weight and 4-bit activation for AlexNet and ResNet50.

- Weighted-entropy-based quantization (WEP, Park et al. (2017)): A quantization scheme that considers statistics of weight/activation. We compared the top-1/top-5 reported accuracy of their quantization with various bit-precision for AlexNet, where the first and last layers are not quantized.

- Low-precision batch normalization (LPBN, Graham (2017)): A scheme for activation quantization in the process of batch normalization. We compared the top-1/top-5 reported accuracy of their quantization with 3-5 bit precision for activation. The first layer activation is not quantized.

- Half-wave Gaussian quantization (HWGQ, Cai et al. (2017)): A quantization scheme that finds the scale via Lloyd search on Normal distribution. We compared the top-1/top-5 reported accuracy for their quantization with 1-bit weight and varying activation bit-precision for AlexNet, and 2-bit weight for ResNet18 and ResNet50. The first and last layers are not quantized.

### E.2 COMPARISON OF ACCURACY

In this section, we present full comparison of accuracy (top-1 and top-5) of the tested CNNs (AlexNet, ResNet18, ResNet50) for image classification on IMAGENET dataset. All the data points for PACT and DoReFa are obtained by running experiments on Tensorpack. All the other data points are accuracy reported in the corresponding papers. As can be seen, PACT achieves the best accuracy across the board for various flavors of quantization. We also observe that using PACT for activation quantization enables more aggressive weight quantization without loss in accuracy.

Table 3: Comparison of Top-1 accuracy (in %) for AlexNet. Bold entries indicate the lowest accuracy degradation compared to single-precision reference from each work. Baseline (full-precision) accuracy for PACT is 55.1%.

| BitW | 32 | 32 | 32 | 32 | 32 | 2 | 2 | 2 | 3 | 3 | 3 | 4 | 4 | 4 | 5 |
|---|---|---|---|---|---|---|---|---|---|---|---|---|---|---|---|
| BitA | 32 | 2 | 3 | 4 | 5 | 2 | 3 | 4 | 2 | 3 | 4 | 2 | 3 | 4 | 5 |
| WRPN | 57.2 | 52.7 | | 54.4 | | 51.3 | | 50.5 | | | | 52.4 | | 54.4 | |
| BalancedQ | 57.1 | 56.5 | | | | 55.7 | | | | | | | | | |
| FGQ | 56.8 | | | | | | | 49.0 | | | | | | | |
| WEQ | 57.1 | | | | | 50.6 | 53.7 | 54.4 | 51.8 | 54.9 | 55.5 | 52.3 | 55.1 | 55.9 | |
| DoReFa | 55.1 | 54.1 | 55.1 | 54.8 | 54.9 | 46.4 | | | | 45.0 | | | | 45.1 | 45.1 |
| PACT | 55.1 | **54.9** | **55.6** | **55.5** | **55.2** | **55.0** | **55.4** | **55.7** | **54.6** | **55.6** | **55.7** | **54.6** | **55.7** | **55.7** | **55.7** |

Table 4: Comparison of Top-5 accuracy (in %) for AlexNet. Bold entries indicate the lowest accuracy degradation compared to single-precision reference from each work. Baseline (full-precision) accuracy for PACT is 77.0%.

| BitW | 32 | 32 | 32 | 32 | 32 | 2 | 2 | 2 | 3 | 3 | 3 | 4 | 4 | 4 | 5 |
|---|---|---|---|---|---|---|---|---|---|---|---|---|---|---|---|
| BitA | 32 | 2 | 3 | 4 | 5 | 2 | 3 | 4 | 2 | 3 | 4 | 2 | 3 | 4 | 5 |
| LogQuant | 78.3 | | 77.1 | | | | | | | | | | | | |
| WEQ | 80.2 | | | | | 75.0 | 77.5 | 78.0 | 76.0 | 78.5 | 79.1 | 76.5 | 78.5 | 79.2 | |
| BalancedQ | 79.4 | 79.0 | | | | 78.0 | | | | | | | | | |
| DoReFa | 77.0 | 76.9 | **77.9** | 77.5 | **77.5** | 76.8 | | | | 77.8 | | | | 77.5 | **77.9** |
| PACT | 77.0 | **77.2** | 77.8 | **77.6** | 77.2 | **77.7** | **77.9** | **78.0** | **77.1** | **78.0** | **78.0** | **77.1** | **78.0** | **78.0** | 77.8 |

Table 5: Comparison of Top-1 accuracy (in %) for ResNet18. Bold entries indicate the lowest accuracy degradation compared to single-precision reference from each work. Baseline (full-precision) accuracy for PACT is 70.4%.

| BitW | 32 | 32 | 32 | 32 | 32 | 1 | 1 | 1 | 1 | 2 | 3 | 4 | 5 |
|---|---|---|---|---|---|---|---|---|---|---|---|---|---|
| BitA | 32 | 2 | 3 | 4 | 5 | 32 | 2 | 3 | 4 | 2 | 3 | 4 | 5 |
| BalancedQ | 68.2 | 62.1 | | | | | | | | 59.4 | | | |
| LPBN | 69.6 | | 63.6 | 66.7 | 69.3 | | | | | | | | |
| HWGQ | 69.6 | | | | | 61.3 | 57.6 | 60.3 | 60.8 | | | | |
| DoReFa | 70.4 | 66.9 | 68.3 | 68.5 | 68.7 | | | | | 62.6 | 67.5 | 68.1 | 68.4 |
| PACT | 70.4 | **67.5** | **69.2** | **70.0** | **70.0** | **65.8** | **62.9** | **65.3** | **65.0** | **64.4** | **68.1** | **69.2** | **69.8** |

Table 6: Comparison of Top-5 accuracy (in %) for ResNet18. Bold entries indicate the lowest accuracy degradation compared to single-precision reference from each work. Baseline (full-precision) accuracy for PACT is 89.6%.

| BitW | 32 | 32 | 32 | 32 | 32 | 1 | 1 | 1 | 1 | 2 | 3 | 4 | 5 |
|---|---|---|---|---|---|---|---|---|---|---|---|---|---|
| BitA | 32 | 2 | 3 | 4 | 5 | 32 | 2 | 3 | 4 | 2 | 3 | 4 | 5 |
| BalancedQ | 87.5 | 82.7 | | | | | | | | 82.0 | | | |
| LPBN | 89.2 | | 85.2 | 87.5 | 88.8 | | | | | | | | |
| HWGQ | 89.2 | | | | | 83.6 | 81.0 | 82.8 | 83.4 | | | | |
| DoReFa | 89.6 | 87.3 | 88.2 | 88.5 | 88.6 | | | | | 84.4 | 87.6 | 88.1 | 88.3 |
| PACT | 89.6 | **87.6** | **88.9** | **89.3** | **89.3** | **86.7** | **84.7** | **85.9** | **85.9** | **85.6** | **88.2** | **89.0** | **89.3** |

Table 7: Comparison of Top-1 accuracy (in %) for ResNet50. Bold entries indicate the lowest accuracy degradation compared to single-precision reference from each work. Baseline (full-precision) accuracy for PACT is 76.9%.

| BitW | 32 | 32 | 32 | 32 | 1 | 2 | 2 | 3 | 4 | 5 |
|---|---|---|---|---|---|---|---|---|---|---|
| BitA | 32 | 3 | 4 | 5 | 2 | 2 | 4 | 3 | 4 | 5 |
| FGQ | 75.1 | | | | | | 68.4 | | | |
| LPBN | 76.0 | 56.1 | 73.8 | 75.6 | | | | | | |
| HWGQ | 76.0 | | | | 64.6 | | | | | |
| DoReFa | 76.9 | | | | | 67.1 | | 69.9 | 71.4 | 71.4 |
| PACT | 76.9 | **75.5** | **75.9** | **76.0** | **67.8** | **72.2** | **74.5** | **75.3** | **76.5** | **76.7** |

Table 8: Comparison of Top-5 accuracy (in %) for ResNet50. Bold entries indicate the lowest accuracy degradation compared to single-precision reference from each work. Baseline (full-precision) accuracy for PACT is 93.1%.

| BitW | 32 | 32 | 32 | 32 | 1 | 2 | 2 | 3 | 4 | 5 |
|---|---|---|---|---|---|---|---|---|---|---|
| BitA | 32 | 3 | 4 | 5 | 2 | 2 | 4 | 3 | 4 | 5 |
| LPBN | 93.0 | 79.6 | 91.8 | 92.6 | | | | | | |
| HWGQ | 93.0 | | | | 85.9 | | | | | |
| DoReFa | 93.1 | | | | | 87.3 | | 89.2 | 89.8 | 93.3 |
| PACT | 93.1 | **92.6** | **92.9** | **92.9** | **87.9** | **90.5** | **91.9** | **92.6** | **93.2** | **93.3** |

