# OpenReview forum: "PACT: Parameterized Clipping Activation for Quantized Neural Networks"
_ICLR.cc/2018/Conference — Reject_

### Official Review · AnonReviewer2 · 2017-11-27
**Is the idea strong enough?**

**Rating:** 5
**Confidence:** 5

**Review:**

The parameterized clipping activation (PACT) idea is very clear: extend clipping activation by learning the clipping parameter. Then,  PACT is combined with quantizing the activations.

The proposed technique sounds. The performance improvement is expected and validated by experiments.

But I am not sure if the novelty is strong enough for an ICLR paper.

---

> ### Author Response · Authors · 2017-12-12
> **PACT UNIVERSALLY outperforms ReLU based activation quantization schemes**
>
> Thank you for your review and showing interest to our work. To answer your question on the novelty of PACT, we put a detail response in the first comment above.
>
> And here's a brief summary: We claim that PACT is a new activation function that is best suitable for activation quantization. We claim that (1) PACT demonstrates (for the first time) no-accuracy-degraded 4-bit quantization (both weight and activation) for challenging ResNet-50 for ImageNet dataset, and (2) PACT UNIVERSALLY outperforms ReLU based activation quantization schemes for all the CNN models we tested.
>
> To better explain why PACT outperforms ReLU based activation quantization schemes, we newly added Appendix A and B for deeper analysis of PACT. We showed that (1) PACT is as expressive as ReLU, and (2) PACT balances clipping and quantization errors when activation is quantized.
>
> Also, please note that all the robust accuracies we achieved with PACT do NOT require any modification in the original hyper-parameters and network structures the baselines use.

---

> ### Author Response · Authors · 2018-01-05
> **Significant portion of new materials are added to support novelty and demonstrate superiority of PACT. Please share your thoughts.**
>
> We thank Reviewer2 for time and effort for reviewing our paper.
>
> We have updated our draft with the following significant changes in contents (all the changes are colored in blue):
>
> 1) We added accuracy comparison between PACT and DoReFa-Net (state-of-the-art in quantized neural network) for ResNet-50 (in Table 1) to demonstrate superiority of PACT. Notice that PACT outperforms DoReFa-Net with > 5% higher accuracy. This confirms that PACT enables no accuracy degradation for the quantized ResNet-50, which was not achievable by previous state-of-the-art activation quantization schemes. Also note that this superior accuracy is achieved without ANY tuning in hyper-parameters; the same hyper-parameters are used in both baseline and the PACT experiments and the networks are trained from scratch. This indicates that Deep Learning practitioners can simply replace ReLU with PACT to achieve robust accuracy when quantizing activation of their neural networks. This claim is supported by our extensive experimental results in Section 5 and Appendix E.
>
> 2) We added two sections in the Appendix to explain why PACT is superior than both ReLU and Clipping activation function for quantized neural networks. In Appendix A, we show theoretical analysis that PACT is as expressive as ReLU when it is used as an activation function. Furthermore,  we explain in Appendix B that PACT finds a balancing point between clipping and quantization errors to minimize their impact to classification accuracy. This analysis demonstrates novelty of PACT as a superior activation function for quantized neural network.
>
> 3) We reflect comments from Reviewer 1 in the draft. We merged the contribution statements in Section 1 (Introduction), and clarified experimental settings to highlight that the baseline networks use ReLU as described in the references, and the PACT experiments use the identical hyper-parameters as the baseline, except that the activation function is replaced from ReLU to PACT.
>
> Please read the updated draft and share your thoughts. Especially, we are curious about in which aspect the reviewer thinks that our paper lacks novelty. Any detail comments would be very appreciated for improving our paper.

---

### Official Review · AnonReviewer1 · 2017-11-28
**This paper proposes to use a clipping activation function as a replacement of ReLu to train a neural network with quantized weights and activations.**

**Rating:** 5
**Confidence:** 4

**Review:**

The authors have addressed my concerns, and clarified a misunderstanding of the baseline that I had, which I appreciate. I do think that it is a solid contribution with thorough experiments. I still keep my original rating of the paper because the method presented is heavily based on previous works, which limits the novelty of the paper. It uses previously proposed clipping activation function for quantization of neural networks, adding a learnable parameter to this function.
_______________
ORIGINAL REVIEW:

This paper proposes to use a clipping activation function as a replacement of ReLu to train a neural network with quantized weights and activations. It shows empirically that even though the clipping activation function obtains a larger training error for full-precision model, it maintains the same error when applying quantization, whereas training with quantized ReLu activation function does not work in practice because it is unbounded.

The experiments are thorough, and report results on many datasets, showing that PACT can reduce down to 4 bits of quantization of weights and activation with a slight loss in accuracy compared to the full-precision model.
Related to that, it seams a bit an over claim to state that the accuracy decrease of quantizing the DNN with PACT in comparison with previous quantization methods is much less because the decrease is smaller or equal than 1%, when competing methods accuracy decrease compared to the full-precision model is more than 1%. Also, it is unfair to compare to the full-precision model using clipping, because ReLu activation function in full-precision is the standard and gives much better results, and this should be the reference accuracy. Also, previous methods take as reference the model with ReLu activation function, so it could be that in absolute value the accuracy performance of competing methods is actually higher than when using PACT for quantizing DNN.

OTHER COMMENTS:

- the list of contributions is a bit strange. It seams that the true contribution is number 1 on the list, which is to introduce the parameter \alpha in the activation function that is learned with back-propagation, which reduces the quantization error with respect to using ReLu as activation function. To provide an analysis of why it works and quantitative results, is part of the same contribution I would say.

---

> ### Author Response · Authors · 2017-12-12
> **Experiments on PACT show a clear trend that it outperforms ReLU based activation quantization schemes**
>
> Thank you for the detail comments. Here are our answers:
>
> Q1. Over-claim that PACT’s accuracy degradation is much less than others?
> A1. There are two aspects to consider for PACT's accuracy degradation.
> First, PACT outperforms (in terms of accuracy degradation) for all the bit-width configuration we compared, demonstrating superior robustness of PACT compared to the other quantization schemes. This clear trend can be seen in Tables 3-8, where the bold numbers indicate the one with lowest accuracy degradation for each column. We added Appendix A and B to analyze why PACT can outperform ReLU based activation quantization schemes.
>
> Second, PACT's accuracy degradation is much lower for the challenging activation quantization (e.g., quantizing activation of binary/ternary weight networks) for ResNet-50. For example, as shown in Table 7, accuracy degradations for HWGQ and FGQ are 11.4% and 6.7%, respectively, whereas PACT's accuracy degradations are 9.1% and 2.4% for the same bit-precision. This gap in accuracy degradation becomes even larger when PACT is compared to the LPBN technique. In case of 3-bit activation with full-precision weight, LPBN's accuracy degradation is 19.9%, whereas PACT's accuracy degradation is only 1.4%.
>
>
> Q2. Baseline uses Clipping activation function?
> A2. No, our full-precision baselines use the same activation function (i.e., ReLU) as the network structure is proposed in the original paper. Tables 3-8 show that the accuracies for our full-precision baselines are comparable to the full-precision reference of the other work we compared. We will clarify this more in Section 5 and Appendix D.
>
>
> Q3. Do not separate contribution for “Why PACT works” with “PACT”
> A3. Thanks for the suggestion. We will merge the first two contributions to one. Furthermore, we now include enhanced analysis on PACT in Appendix A and B to provide deeper understanding about why PACT outperforms previous ReLU based activation quantization schemes.

---

> ### Author Response · Authors · 2018-01-05
> **Significant portion of new materials are added to support novelty and demonstrate superiority of PACT. Please share your thoughts.**
>
> We thank Reviewer1 for time and effort for reviewing our paper.
>
> We have updated our draft with the following significant changes in contents (all the changes are colored in blue):
>
> 1) We reflect your comments in the draft. We merged the contribution statements in Section 1 (Introduction), and clarified experimental settings to highlight that the baseline networks use ReLU as described in the references, and the PACT experiments use the identical hyper-parameters as the baseline, except that the activation function is replaced from ReLU to PACT.
>
> 2) We added accuracy comparison between PACT and DoReFa-Net (state-of-the-art in quantized neural network) for ResNet-50 (in Table 1) to demonstrate superiority of PACT. Notice that PACT outperforms DoReFa-Net with > 5% higher accuracy. This confirms that PACT enables no accuracy degradation for the quantized ResNet-50, which was not achievable by previous state-of-the-art activation quantization schemes. Also note that this superior accuracy is achieved without ANY tuning in hyper-parameters; the same hyper-parameters are used in both baseline and the PACT experiments and the networks are trained from scratch. This indicates that Deep Learning practitioners can simply replace ReLU with PACT to achieve robust accuracy when quantizing activation of their neural networks. This claim is supported by our extensive experimental results in Section 5 and Appendix E.
>
> 3) We added two sections in the Appendix to explain why PACT is superior than both ReLU and Clipping activation function for quantized neural networks. In Appendix A, we show theoretical analysis that PACT is as expressive as ReLU when it is used as an activation function. Furthermore,  we explain in Appendix B that PACT finds a balancing point between clipping and quantization errors to minimize their impact to classification accuracy. This analysis demonstrates novelty of PACT as a superior activation function for quantized neural network.
>
> Please read the updated draft and share your thoughts. Any comments would be very appreciated for improving our paper.

---

### Official Review · AnonReviewer3 · 2017-12-01

**Rating:** 5
**Confidence:** 4

**Review:**

This paper presents a new idea to use PACT to quantize networks, and showed improved compression and comparable accuracy to the original network. The idea is interesting and novel that PACT has not been applied to compressing networks in the past. The results from this paper is also promising that it showed convincing compression results.

The experiments in this paper is also solid and has done extensive experiments on state of the art datasets and networks. Results look promising too.

Overall the paper is a descent one, but with limited novelty. I am a weak reject

---

> ### Author Response · Authors · 2017-12-12
> **PACT UNIVERSALLY outperforms ReLU based activation quantization schemes**
>
> Thank you for your review and showing interest to our work. To answer your question on the novelty of PACT, we put a detail response in the first comment above.
>
> And here's a brief summary: We claim that PACT is a new activation function that is best suitable for activation quantization. We claim that (1) PACT demonstrates (for the first time) no-accuracy-degraded 4-bit quantization (both weight and activation) for challenging ResNet-50 for ImageNet dataset, and (2) PACT UNIVERSALLY outperforms ReLU based activation quantization schemes for all the CNN models we tested.
>
> To better explain why PACT outperforms ReLU based activation quantization schemes, we newly added Appendix A and B for deeper analysis of PACT. We showed that (1) PACT is as expressive as ReLU, and (2) PACT balances clipping and quantization errors when activation is quantized.
>
> Also, please note that all the robust accuracies we achieved with PACT do NOT require any modification in the original hyper-parameters and network structures the baselines use.

---

> ### Author Response · Authors · 2018-01-05
> **Significant portion of new materials are added to support novelty and demonstrate superiority of PACT. Please share your thoughts.**
>
> We thank Reviewer3 for time and effort for reviewing our paper.
>
> We have updated our draft with the following significant changes in contents (all the changes are colored in blue):
>
> 1) We added accuracy comparison between PACT and DoReFa-Net (state-of-the-art in quantized neural network) for ResNet-50 (in Table 1) to demonstrate superiority of PACT. Notice that PACT outperforms DoReFa-Net with > 5% higher accuracy. This confirms that PACT enables no accuracy degradation for the quantized ResNet-50, which was not achievable by previous state-of-the-art activation quantization schemes. Also note that this superior accuracy is achieved without ANY tuning in hyper-parameters; the same hyper-parameters are used in both baseline and the PACT experiments and the networks are trained from scratch. This indicates that Deep Learning practitioners can simply replace ReLU with PACT to achieve robust accuracy when quantizing activation of their neural networks. This claim is supported by our extensive experimental results in Section 5 and Appendix E.
>
> 2) We added two sections in the Appendix to explain why PACT is superior than both ReLU and Clipping activation function for quantized neural networks. In Appendix A, we show theoretical analysis that PACT is as expressive as ReLU when it is used as an activation function. Furthermore,  we explain in Appendix B that PACT finds a balancing point between clipping and quantization errors to minimize their impact to classification accuracy. This analysis demonstrates novelty of PACT as a superior activation function for quantized neural network.
>
> 3) We reflect comments from Reviewer 1 in the draft. We merged the contribution statements in Section 1 (Introduction), and clarified experimental settings to highlight that the baseline networks use ReLU as described in the references, and the PACT experiments use the identical hyper-parameters as the baseline, except that the activation function is replaced from ReLU to PACT.
>
> Please read the updated draft and share your thoughts. Especially, we are curious about in which aspect the reviewer thinks that our paper lacks novelty. Any detail comments would be very appreciated for improving our paper.

---

### Author Response · Authors · 2017-12-12
**Response to the question about novelty of PACT**

The authors thank reviewers for their contribution to improving this paper.

We'd like to highlight the following 3 novel aspects of our PACT paper and we're hoping this communicates to the reviewers the significance of our work:

(A) NO loss of accuracy during quantization:  Over the past 3 years, there has been a tremendous amount of work focused on quantization (binarization / ternarization etc.) of neural networks. Most of these publications focused on applying these techniques to simpler networks (based on the CIFAR10, SVHN and MNIST datasets) where they reported little loss of accuracy. However, in cases when the same exact techniques were applied to larger models (based on the ImageNet dataset), significant loss of accuracy has been reported - leading us to conclude that all previous quantization techniques ONLY work when there is significant redundancy in the model and do not scale well to state of the art networks.

As Table 7 shows, our work is the first paper that shows state-of-the-art accuracy (<0.5% Top-1 accuracy degradation and slight IMPROVEMENT in Top-5 accuracy) using 4-bit quantizations for both weights and activations for ResNet-50 for ImageNet dataset. Furthermore, our work shows robust accuracy for ternary and binary weight network (with 4 and 2-bit activation, respectively), when all previous techniques showed significantly (2.3 ~ 4.2% Top-1) more accuracy degradation.

This is critically important since a significant number of models in Medical Imaging [1], Automotive [2] and other domains are based on transfer learning applied to ResNet like models (based on ImageNet) - and preserving accuracy is extremely critical in these domains due to its direct impact on safety and human life. Allowing 4-bit quantizations to work with the same level of accuracy, as discussed in depth in this paper (Section 6), also allows 2X improvement in Inference/Watt throughput in co./mparison to state-of-the-art 8-bit models - which is critical for power-constrained mobile, IoT and even Cloud hardware devices.
[1] Litjens, Geert, et al. "A survey on deep learning in medical image analysis." arXiv preprint arXiv:1702.05747 (2017).
[2] Adam Grzywaczewski, “Training AI for Self-Driving Vehicles: the Challenge of Scale,” in https://devblogs.nvidia.com/parallelforall/training-self-driving-vehicles-challenge-scale/


(B) Second, PACT shows superior performance, quantified in terms of accuracy degradation, for all the bit-configurations and networks tested in comparison to seven state-of-the-art quantization publications. From Tables 3-8, we can observe that the accuracy degradation (averaged over all the bit-configurations) of the compared publications for AlexNet, ResNet18, and ResNet50 are 6.1%, 5.1% and 8.1% (Top-1), respectively. In contrast, PACT's accuracy degradation (averaged across all the bit-configurations) is -0.2% (i.e., achieving slightly better accuracy than reference), 3.1% and 2.7%. (Tables 3-8 also highlight in bold which scheme achieves the lowest accuracy degradation for each bit-configuration.)

This showcases the reliability of PACT for quantization. Please note that large degradation in accuracy nullifies the use of large scale DNNs - rendering previous techniques largely unusable in most scenarios. For example, ResNet-18 takes 3.6B Flops to achieve 72.12% Top-1 accuracy, whereas ResNet-50 takes 7.6B Flops to achieve 77.15% Top-1 accuracy. Thus it's better to use ResNet-18 if accuracy degradation is >3% for ResNet-50.


(C) Third, PACT has a unique characteristic that balances clipping and quantization errors when quantizing activation. We newly added a deeper analysis on why PACT outperforms other activation functions for activation quantization in Appendix A and B. In Appendix A, we showed the expressivity of PACT that it can be trained via SGD to tune the clipping levels properly in order to produce the output that the same network with ReLU would produce. This tuning capability was validated with the CIFAR10-ResNet20 experiment shown in Fig. 7 that PACT based networks could converge to almost identical training curves in comparison to the ReLU based network.

In Appendix B, we further explained why PACT provides robustness to activation quantization. We first observed that when activation is quantized, there is a trade-off between clipping and quantization errors depending on the clipping level (Fig. 8a). We demonstrated that PACT auto-tunes the clipping level during training to achieve optimal accuracy under activation quantization (Fig. 8b). Since PACT does not require sweeping to obtain the right clipping level, this is a very computationally feasible way.

---

> ### Author Response · Authors · 2017-12-12
> **(Cont'd) PACT needs NO change in hyper-parameters from baseline**
>
> Furthermore, we want to emphasize that PACT's robust accuracy is achieved WITHOUT any changes to the original model, except that ReLU is replaced with PACT. In other words, we used the same hyper-parameters (learning rate schedules, weight initialization, mini batch size, optimizers (ADAM or SGD with momentum), etc.) as well as the original models and network structures in all of our experiments. Furthermore, all of the training was done from scratch, showing that this work does not require any pre-trained weights for good initialization, or any warm-start or larger number of training epochs.

---

### Decision · Program_Chairs · 2018-01-29
**ICLR 2018 Conference Acceptance Decision**

**Decision:**

Reject

**Comment:**

All of the reviewers agree that the experimental results are promising and the proposed activation function enables a decent degree of quantization. However, the main concern with the approach is its limited novelty compared to previous work on clipped activation functions.

minor comments:
- Even though PACT is very similar to Relu, the names are very different.
- Please include a plot showing the proposed activation function as well.